# Identification of a Novel Protein-Based Prognostic Model in Gastric Cancers

**DOI:** 10.3390/biomedicines11030983

**Published:** 2023-03-22

**Authors:** Zhijuan Xiong, Chutian Xing, Ping Zhang, Yunlian Diao, Chenxi Guang, Ying Ying, Wei Zhang

**Affiliations:** 1Department of Gastroenterology, The First Affiliated Hospital of Nanchang University, Nanchang 330006, China; 2Jiangxi Medical Center for Major Public Health Events, The First Affiliated Hospital of Nanchang University, Nanchang 330006, China; 3The Department of Respiratory and Intensive Care Medicine, The First Affiliated Hospital of Nanchang University, Nanchang 330006, China; 4Queen Mary School, Nanchang University, Nanchang 330006, China

**Keywords:** gastric cancer, TCPA, overall survival, TIGAR

## Abstract

Gastric cancer (GC) is the third leading cause of cancer-related deaths worldwide. However, there are still no reliable biomarkers for the prognosis of this disease. This study aims to construct a robust protein-based prognostic prediction model for GC patients. The protein expression data and clinical information of GC patients were downloaded from the TCPA and TCGA databases, and the expressions of 218 proteins in 352 GC patients were analyzed using bioinformatics methods. Additionally, Kaplan–Meier (KM) survival analysis and univariate and multivariate Cox regression analysis were applied to screen the prognosis-related proteins for establishing the prognostic prediction risk model. Finally, five proteins, including NDRG1_pT346, SYK, P90RSK, TIGAR, and XBP1, were related to the risk prognosis of gastric cancer and were selected for model construction. Furthermore, a significant trend toward worse survival was found in the high-risk group (*p* = 1.495 × $10^{-7}$). The time-dependent ROC analysis indicated that the model had better specificity and sensitivity compared to the clinical features at 1, 2, and 3 years (AUC = 0.685, 0.673, and 0.665, respectively). Notably, the independent prognostic analysis results revealed that the model was an independent prognostic factor for GC patients. In conclusion, the robust protein-based model based on five proteins was established, and its potential benefits in the prognostic prediction of GC patients were demonstrated.

## 1. Introduction

Gastric cancer is a major global health concern and remains the fifth most common malignant tumor, with more than 1 million new cases every year. Additionally, gastric cancer is the third leading cause of cancer-related deaths with an estimated 0.8 million new deaths worldwide in 2020 [1]. The most prevalent histological subtype of GC is the adenocarcinoma because it accounts for more than 95% of GC cases [2]. Despite great progress in the earlier diagnosis of and better treatment for GC in the past few decades, still, most patients have already metastasized or progressed to higher malignancy levels once diagnosed and have a poor prognosis. Furthermore, it is reported that the 5-year survival rate of GC patients was as low as 31% in the United States [3,4]. To obtain optimal treatment strategies for each patient, clinicians need to recognize the risk for GC patients according to the prognoses by using reliable and predictive biomarkers. Hence, it is urgent and imperative to develop a robust model for early diagnosis and improving the prediction of prognosis for GC.

With the development of next-generation sequencing (NGS) technologies, more and more studies primarily focused on searching for reliable biomarkers for better predicting the prognosis of GC patients, including using large mRNA prognostic markers or models to predict the prognosis of GC according to the TCGA and GEO databases [5,6,7]. However, the clinical application of NGS is limited due to its high cost, and unsuitability for high-throughput application and data analysis. Recently, protein-based prognostic risk models have gained more and more attention because the detection of tumor protein biomarkers is economical, reliable, reproducible, and more accessible in clinical practice. It is reported that protein-based prognostic models have been successfully constructed and validated predictive accuracy in patients with breast carcinoma and prostate carcinoma [8,9]. To date, the protein-based prognostic signature has rarely been used to predict the prognosis of GC patients. Therefore, it is crucial to establish an effective risk model based on protein for GC patients.

Reverse-phase protein arrays (RPPAs) are a powerful proteomic tool that can detect the species and quantity of interesting proteins in multiple samples in a high-throughput manner [10]. The Cancer Proteome Atlas (TCPA) is an open database with many tumor protein profile data, which come from integrating the RPPA chip data from The Cancer Genome Atlas (TCGA) and several independent oncology research programs; it contains nearly 300 protein expression data in more than 8000 tumor samples from 32 cancer types in the TCGA [11,12]. Therefore, it is feasible to construct a protein prognostic risk model for GC based on TCPA and TCGA databases.

In this study, we aimed to build a reliable protein-based risk model that might contribute to predicting prognosis for GC patients by using data from TCPA and TCGA databases. Through a comprehensive bioinformatics analysis, we constructed a new five-protein prognostic risk model that outperformed traditional clinical features in predicting overall survival in GC patients. Furthermore, this model had a higher specificity and sensitivity compared to the clinical features and could be used as an independent prognostic factor for GC patients. Our findings could contribute to the development of personalized treatment strategies and improve the clinical management of GC patients.

## 2. Materials and Methods

### 2.1. Data Collection and Processing

To establish a reliable protein-based risk model for GC patients, we first downloaded the protein expression data of GC patients (level 4 data) from the TCPA database (https://tcpaportal.org/tcpa/ (accessed on 5 March 2022)), which included 392 tumor samples and the protein expression of 218 proteins. Then, we also obtained the matching clinical information of GC patients from the TCGA database (https://gdc.cancer.gov/ (accessed on 8 March 2022)) for further analysis. The clinical information included the general information (such as, age and sex), clinical-pathological data (such as, tumor grade, tumor stage, T stage, N stage and M stage), and the follow-up information (survival status and survival time). The imputation of missing protein expression data was performed using R software, with the “impute” package. The Perl software was used to integrate the protein expression data and corresponding clinical data.

### 2.2. Identification of Prognostic-Related Proteins

In order to identify the proteins that may affect the GC overall survival, we searched for the differential proteins by using a *t*-test of two independent samples in the GC patients and the criterion for differential proteins was the *p*-value < 0.05, a total of 34 differentially expressed proteins were screened. In addition, we simultaneously performed Kaplan–Meier (KM) survival analysis and univariate Cox regression among these proteins using R software with the “survival” package. The proteins with *p*-values < 0.05 for both analyses were identified as prognostic-related proteins.

### 2.3. Construction and Evaluation of Protein Risk Prognostic Model

To identify the risk prognostic proteins, all significant proteins on both the KM and univariate Cox analyses were further performed using multivariable Cox regression analysis, which was a regression analysis methodology that utilized regression analysis for variable selection and regularization to improve the predictive accuracy and interpretability of the statistical model. It was commonly applied to high-dimensional data with low correlation between features to avoid overfitting and achieve optimal feature selection. This approach efficiently identified the most informative predictive markers, facilitating the generation of a prognostic signature for predicting clinical outcomes. Ultimately, five proteins were reserved in the risk prognostic model and their coefficients were retained.

A risk score for each patient was calculated by multiplying the expression level of the model protein and the corresponding regression coefficient. The computed formula was as follows: risk score = (−0.201 × SYK expression) + (−0.298 × NDRG1_pT346 expression) + (−0.489 × P90RSK expression) + (−0.304 × TIGAR expression) + (0.749 × XBP1 expression). The GC patients were divided into a high-risk group and a low-risk group according to the median value of risk scores.

The relationship of each protein expression in the model, survival status, survival time, and risk score values were analyzed and visualized using R software with the “survminer” and “survival” packages. The predictive effect between the high- and low-risk groups was conducted using a log-rank test with KM survival analysis. To verify whether the risk score was an independent prognostic factor for the overall survival of GC patients, univariate and multivariate Cox analyses were performed. The R package “time ROC” was used to draw the time-dependent ROC curve to assess the predictive accuracy of risk score and clinical features.

### 2.4. Co-Expressed Proteins Analysis

The co-expressed proteins typically have similar functions to some extent. To better understand the pathogenic mechanism of the proteins in the model among GC, the Pearson correlation analysis was carried out for searching the co-expressed proteins with the proteins in the model using “ggalluvial” and “ggplot2” R packages. The hazard ratio (HR) of protein less than 1 was considered as a low-risk protein, while HR > 1 was considered as a high-risk protein. The screened criteria included the correlation index > 0.5 and the *p*-value < 0.001.

### 2.5. Statistical Analysis

R software or Perl software was used for analyses and charts visualized in this study. Continuous data were presented as mean ± SD and categorized data were presented as frequency (n) and proportion (%). The differences in protein expression were analyzed using a *t*-test of two independent samples. The Kaplan–Meier (KM) method was used for survival analysis. The Cox regression was used for model construction and independent prognostic analysis. The Pearson correlation was used for correlation analysis. The time-dependent ROC curve was used for assessing the predictive accuracy of the risk model. All the statistical analyses were assessed by R software (R 3.6.2). A *p*-value < 0.05 was statistically significant.

## 3. Results

### 3.1. Baseline Characteristics of Patients

In order to construct the protein-based risk model, we first downloaded the data of protein expression and clinical information of GC patients from a different database. A total of 218 protein expression data in 392 GC tumor samples were obtained from the TCPA database, and a total of 443 GC patients’ information was downloaded from the TCGA database. Then, we integrated the data of protein expression and the clinical information by using R software and Perl software to remove the missing and unknown data. We found that only 352 GC patients had intact clinical information including survival and protein expression data after removing the missing and unknown data on survival time and status. Finally, 352 patients were included in this study and were further used for protein-based risk model construction.

Subsequently, we analyzed the baseline characteristics of GC patients. The clinical characteristics of the GC cohort was presented in Appendix A. The pathological type of GC patients was stomach adenocarcinoma (STAD). Among the GC patients, 222 patients were male (63.1%) and 130 patients were female (36.9%). The average age of the cohort was 65.25 years old. In terms of the pathologic stage, 39 patients were stage I (11.1%), 106 patients were stage II (30.1%), 167 patients were stage III (47.4%), and 40 patients were stage IV (11.4%). In terms of the T stage, 12 patients were T1 (3.4%), 69 patients were T2 (19.6%), 169 patients were T3 (48%), and 102 patients were T4 (29%). In terms of the N stage, 104 patients were N0 (29.6%) and 248 patients were N+ (70.4%). In terms of the M stage, 325 patients were M0 (92.3%) and 27 patients were M1 (7.7%). In terms of tumor grade, 7 patients were G1 (2%), 119 patients were G2 (33.8%), and 226 patients were G3 (64.2%).

### 3.2. Identification of Prognostic-Related Proteins

To identify the prognostic-related proteins, we screened the candidate protein based on the expression of 218 proteins in 352 GC patients and the patients’ survival information by using both KM survival analysis and univariate Cox analysis. After the screening, a total of 15 proteins, including X4EBP1_pS65, CKIT, CAVEOLIN1, COLLAGENVI, CD20, MYH11, RAPTOR, XBP1, CLAUDIN7, EIF4E, SYK, FASN, NDRG1_pT346, P90RSK, and TIGAR, were identified for both analyses that were significantly related to the overall survival of GC patients (Table 1). Eight proteins, including X4EBP1_pS65, CKIT, CAVEOLIN1, COLLAGENVI, CD20, MYH11, RAPTOR, and XBP1 were considered as the high-risk proteins (HR > 1), indicating that their high expression was associated with increased risk of death of GC patients. Seven proteins, including CLAUDIN7, EIF4E, SYK, FASN, NDRG1_pT346, P90RSK, and TIGAR were regarded as protective proteins (HR < 1), implying that their high expression was associated with decreased risk of death of GC patients.

### 3.3. Construction of Protein Risk Prognostic Model

Then, we further constructed the protein risk prognostic model based on the obtained 15 prognostic-related proteins by using multivariate Cox regression analysis. Interestingly, we found that only five proteins, including SYK, NDRG1_pT346, P90RSK, TIGAR, and XBP1, were proved in the risk model (Table 2). As shown in Figure 1, the low SYK expression (*p* = 2.895 × $10^{-3}$, Figure 1A), NDRG1_pT346 (*p* = 3.872 × $10^{-2}$, Figure 1B), P90RSK (*p* = 1.991 × $10^{-3}$, Figure 1C), and TIGAR (*p* = 2.885 × $10^{-4}$, Figure 1D) and the high XBP1 expression (*p* = 8.621 × $10^{-4}$, Figure 1E) were significantly positively associated with the poor overall survival of STAD (stomach adenocarcinoma) patients.

Subsequently, a risk score for each patient was calculated as the sum of each protein’s score by integrating the expression level and regression coefficient of each prognostic protein (Risk score = ($\overset{N}{\underset{i=1}{\sum }}Expi*Coei$). And then, the patients were ranked from small to large values according to the risk score and were subdivided into high-risk and low-risk groups (176 patients in each group) according to the median risk score of each patient (Figure 2A). Notably, we observed that there were more deaths in the high-risk group compared to the low-risk group (Figure 2B). Meanwhile, the expressions of SYK, NDRG1_pT346, P90RSK, TIGAR, and XBP1 are displayed in a risk heatmap in Figure 2C.

### 3.4. Evaluation of Five Protein Risk Prognostic Model

Furthermore, we analyzed the survival of the patients in the high- and low-risk groups by using the Kaplan–Meier survival curve and found that the patients in the high-risk group exhibited an apparent trend in the low survival rate (*p* = 1.495 × $10^{-7}$, Figure 3A). In addition, we used time-dependent ROC analysis to further confirm the predictive risk model’s performance. As shown in Figure 3B, the risk model’ AUC was assessed at 1, 2, and 3 years (AUC = 0.685, 0.673, and 0.665), suggesting that the prognostic model can effectively predict the survival of GC patients. The further time-dependent AUC value of the risk score was higher than the age, gender, grade, stage, T, M, and N values at 1, 2, 3 years (Figure 3C,E). These findings also indicated that the predictive risk model has a higher accuracy and sensitivity compared to the clinical features. Notably, we further determine whether the risk value and other clinical characteristics are the independent prognostic factors using the univariate (Figure 3F) and multivariate Cox (Figure 3G) analysis. We observed that the age (HR = 1.032, 95% CI = 1.012–1.052, *p* = 0.002) and risk model (HR = 2.173, 95% CI = 1.656–2.853, *p* < 0.001) were the independent prognostic factors among the STAD patients.

### 3.5. Co-Expressed Proteins Analysis

In order to extract more biologically meaningful information of the five obtained proteins, we investigated their correlation network by using protein co-expression analysis. The included co-expressed proteins were selected if they had a correlation index > 0.5 and a *p*-value < 0.001. As shown in Figure 4, we found that NDRG1_pT346 exhibited a significantly positive correlation with X4EBP1_pT37T46, AKT_pS473, GSK3ALPHABETA_pS21S9, SRC_pY527, and GSK3_pS9. Additionally, we observed that the TIGAR had a significant positive correlation with CDK1 and CLAUDIN7. These findings indicated that the expression of NDRG1_pT346 and TIGAR might associate with the signaling of X4EBP1_pT37T46, AKT_pS473, GSK3ALPHABETA_pS21S9, SRC_pY527, and GSK3_pS9, CDK1, and CLAUDIN7.

## 4. Discussion

In this study, we obtained the protein expression data and clinical information from the TCPA and TCGA databases and analyzed them using bioinformatics analysis. Finally, we identified five effective protein signatures of GC prognoses, including NDRG1_pT346, SYK, P90RSK, TIGAR, and XBP1. According to the expression of these five proteins, a gastric cancer prediction protein risk prognostic model was successfully established. Additionally, a GC patient can be classified into a high- or low-risk group based on the protein risk prognostic mode. Notably, we found that the patients in the high-risk group had poorer prognoses. The results of the time-dependent ROC and the univariate and multivariate Cox analysis indicated that the risk prognostic model had a higher accuracy and sensitivity compared to the clinical features and can be used as an independent prognostic factor of gastric cancer.

In the five proteins we identified, studies have demonstrated that the expression of these proteins is associated with gastric cancer development and progression. Among them, NDRG1, SYK, and P90RSK are downregulated and suggest a poor prognosis. NDRG1 is a member of the NDRG family, which was involved in a variety of physiological functions such as cell differentiation [13], stress response [14], and apoptosis [15]. The RNA-seq data of gastric cancer from the TCGA database demonstrated that the mRNA expression of NDRG1 was downregulated in STAD tumor tissues and was significantly negatively correlated with invasion depth [16]. NDRG1 overexpression could significantly suppress the proliferation and invasion and could induce a G1 cell cycle arrest in AGS cells [17,18]. These findings suggested that the protein expression of NDRG1 was downregulated in gastric tumor tissues and related to poor prognosis in GC patients.

Similarly, SYK, a cytosolic non-receptor protein tyrosine kinase, was downregulated and significantly associated with malignancy features such as tumor depth, lymphatic invasion, venous invasion, and lymph node metastasis [19]. In addition, the downregulated SYK expression suggested a significantly lower 5-year survival rate [19]. The hypermethylation of the SYK promoter was positively correlated with tumorigenesis and metastasis in gastric cancer, highly indicating that SYK is a potential tumor suppressor in gastric cancer [20].

The P90RSK, also known as RSK1, is a member of the 90 kDa ribosomal s6 kinases (RSKs) family consisting of four isoforms (RSK1-4) [21]. However, its expression is varied in different tumors. It has been reported that the expression of P90RSK is highly expressed in colon cancer [22], but is downregulated in breast cancer [23]. Similarly, we found that the expression of P90RSK was downregulated in this study, highly suggested that further studies are needed for investigating the role of P90RSK in the development of cancer.

The TIGAR (TP53-induced glycolysis and apoptosis regulator) is the downstream target protein of p53, which plays a critical role in glycolysis and redox balance [24]. The TIGAR is highly expressed in a variety of solid tumors, such as lung cancer [25], colon cancer [26], breast cancer, and gastric cancer [27]. A study from China found that the TIGAR expression was increased in GC and that TIGAR downregulation inhibited cell proliferation and induced cell apoptosis. The underlying mechanism indicates that the TIGAR could protect malignant cells from oxidative stress-caused damage and inhibit the glycolysis process. Therefore, the patients with a high TIGAR expression had a lower 5-year overall survival rate [28]. However, in our study, the patients with higher TIGAR protein expression were associated with the low-risk group and had a higher overall survival rate, which was inconsistent with the available evidence. A potential reason for the contrast was that the patients of the cohort were all white. Currently, few studies focus on the function of the TIGAR in gastric cancer; thus, future studies need to confirm the relationship of TIGAR with GC prognoses.

X-box binding protein 1 (XBP1) is a unique basic-region leucine zipper (bZIP) transcription factor that is involved in regulating the expression of the MHC class II gene and the unfolded protein response (UPR) in endoplasmic reticulum (ER) stress [29]. XBP1 exerts various functions for cancer development, such as promoting cell proliferation, migration, and invasion, and inhibits cell apoptosis or autophagy [30]. It was highly expressed in some solid tumors and is associated with poor prognoses, such as breast cancer [31], lung cancer [32], and liver cancer [33]. However, the role of XBP1 in gastric cancer remains unknown and needs to be further explored.

Furthermore, we also analyzed the co-expressions of NDRG1_pT346, SYK, P90RSK, TIGAR, and XBP1. Performing protein co-expression analysis could provide a complementary view of the underlying mechanism of the target protein in diseases. Interestingly, we found that NDRG1_pT346 was significantly related with X4EBP1_pT37T46, AKT_pS473, GSK3ALPHABETA_pS21S9, SRC_pY527, and GSK3_pS9 and that the TIGAR was remarkably related to CDK1 and CLAUDIN7. X4EBP1_pT37T46, AKT_pS473, GSK3ALPHABETA_pS21S9, and GSK3_pS9 are involved in PI3-AKT signaling, implying that NDRG1_pT346 may exhibit a function via regulating PI3-AKT signaling. CDK1 is a key regulator of the cell cycle [34], and CLAUDIN7 is involved in the process of migration [35]. These findings suggested that in the risk model, the TIGAR and NDRG1 played key roles in the occurrence and development of GC, which were associated with cell cycle, proliferation, and migration.

In conclusion, our study focused on protein profile changes in GC patients and successfully constructed a protein risk prognostic model, which has a potential application for STAD patients. Our findings shared some similarities with Zhang et al. [36], in which a tri-protein prognostic risk model was constructed and used as an independent prognostic factor for stomach adenocarcinoma patients. Compared with Zhang et al. [36], the sample size that we used for model construction was doubled, which is conducive to the credibility of the model. In addition, our evaluation of the model was more reliable as we used time-dependent ROC to assess the prognostic predictive accuracy of the risk model. However, the limitation of this study is the lack of clinical samples to verify the expression of the five proteins and relevant studies. Further investigations are needed to confirm the feasibility of this model.

## Figures and Tables

**Figure 1 biomedicines-11-00983-f001:**
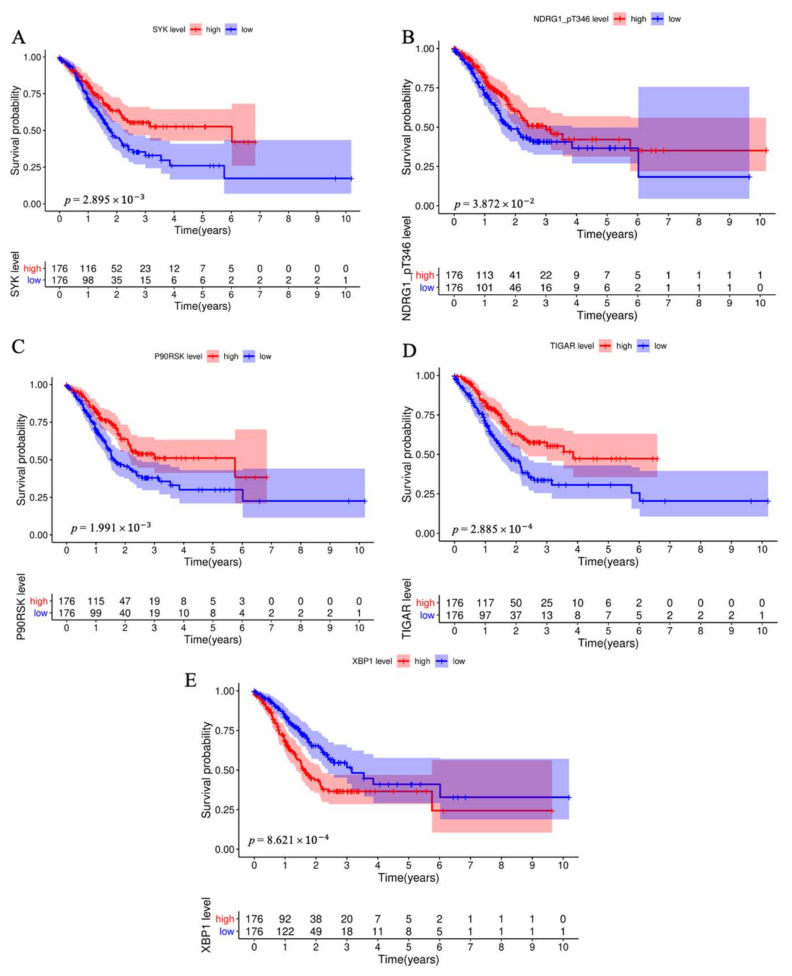
Survival analysis for each model protein. Low SYK expression (**A**), NDRG1-PT346 (**B**), P90RSK (**C**), and TIGAR (**D**) and high XBP1 expression (**E**) were positively associated with poorer overall survival of STAD patients.

**Figure 2 biomedicines-11-00983-f002:**
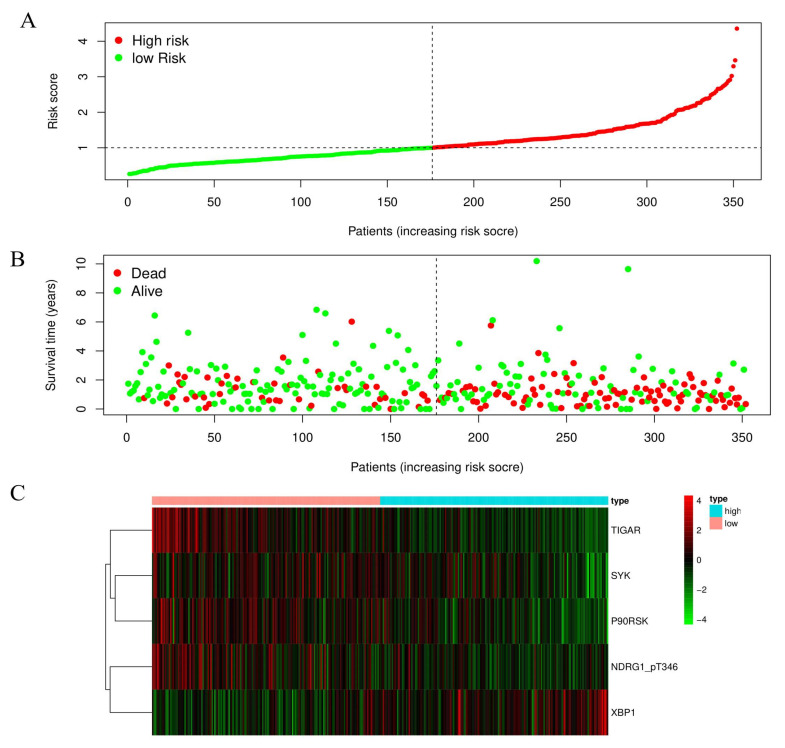
Construction of a protein risk prognostic model in STAD. (**A**) The patients were divided into high-risk and low-risk groups based on the risk scores. The risk scores for all patients were ranked in ascending order and were divided by the threshold (vertical dotted line). The green dots in the left and red dots in the right indicated the low- and high-risk groups respectively. (**B**) The scatter diagram showed the survival time and survival status of each patient in high- and low-risk groups. The red and green dots represented death and alive respectively. The high-risk group had more deaths compared to low-risk group. (**C**) The risk heatmap showed the expression levels of five prognostic-risk model proteins between high-risk and low-risk groups. The dark red and green represented higher expression and lower expression respectively.

**Figure 3 biomedicines-11-00983-f003:**
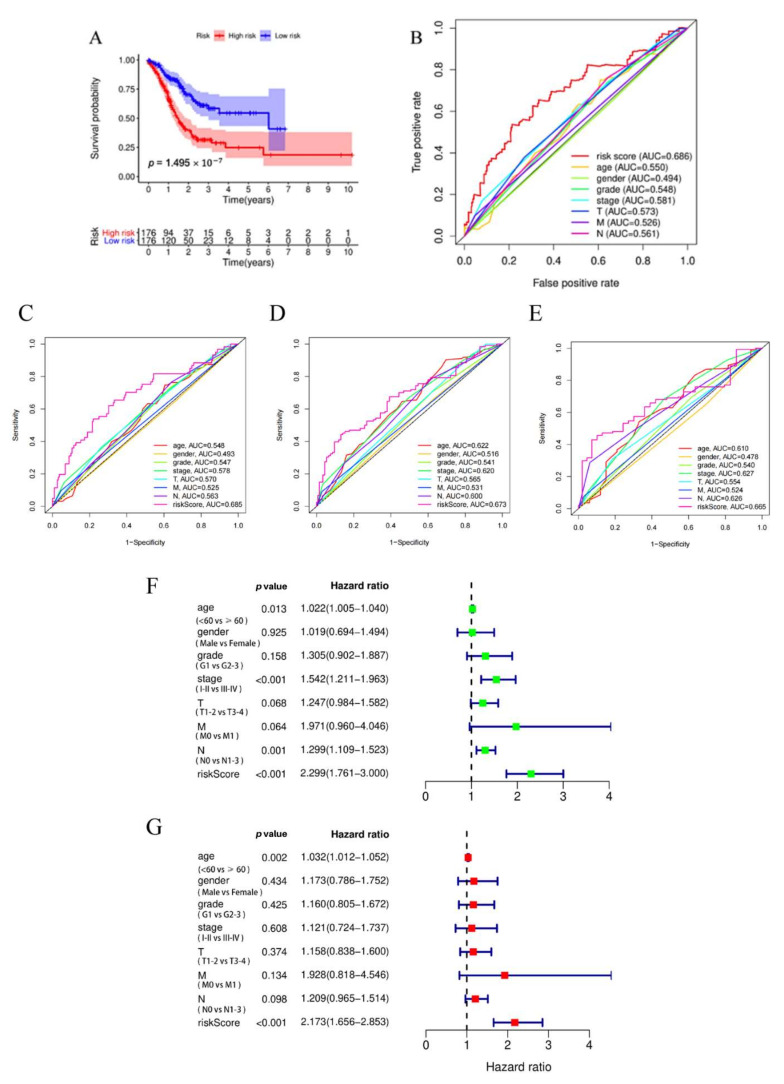
The protein risk prognostic model had good prognostic predictive value and was an independent prognostic factor for STAD patients. (**A**) The survival analysis for STAD patients between high-risk and low-risk groups. (**B**) Time-dependent ROC of the risk model; the AUC was assessed at 1, 2, and 3 years (AUC = 0.685, 0.673, and 0.665). (**C**–**E**) Time-dependent ROC curve analysis revealed that the predictive model has higher accuracy and sensitivity compared to the clinical features. (**F**,**G**) The univariate and multivariate Cox analysis revealed that this risk model was an independent prognostic factor for STAD patients.

**Figure 4 biomedicines-11-00983-f004:**
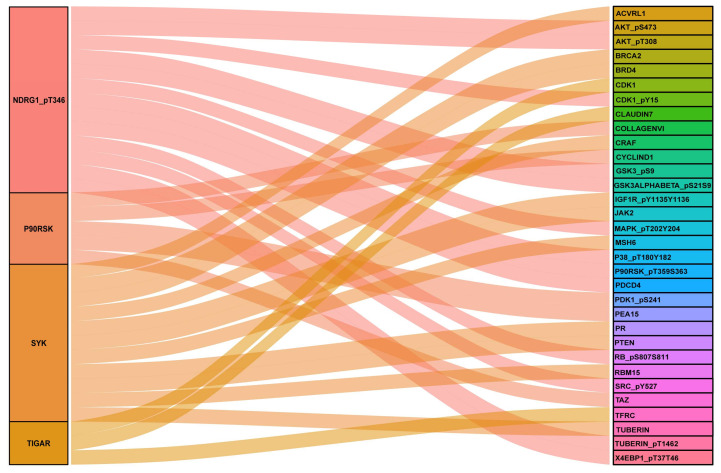
Correlation analyses between model proteins and other proteins based on the TCPA database.

**Table 1 biomedicines-11-00983-t001:** Prognosis-related proteins identified using both Kaplan–Meier (*p* value) survival analysis and univariate Cox analysis.

Protein	Kaplan–Meier	HR	HR.95L	HR.95H	*p* Value
SYK	0.003	0.698	0.537	0.908	0.007
NDRG1_pT346	0.039	0.772	0.599	0.995	0.045
P90RSK	0.002	0.536	0.354	0.812	0.003
TIGAR	0.000	0.695	0.569	0.849	0.000
XBP1	0.001	2.365	1.199	4.667	0.013
X4EBP1_pS65	0.004	1.531	1.042	2.249	0.030
CKIT	0.033	1.410	1.011	1.967	0.043
CAVEOLIN1	0.029	1.267	1.087	1.477	0.002
COLLAGENVI	0.015	1.850	1.225	2.792	0.003
CD20	0.005	1.378	1.089	1.745	0.008
MYH11	0.033	1.069	1.019	1.121	0.006
RAPTOR	0.037	2.124	1.099	4.104	0.025
CLAUDIN7	0.000	0.755	0.652	0.874	0.000
EIF4E	0.041	0.481	0.241	0.962	0.039
FASN	0.003	0.782	0.628	0.974	0.028

**Table 2 biomedicines-11-00983-t002:** Proteins used for the construction of risk prognostic model.

Protein	Coef	HR	HR.95L	HR.95H	*p* Value
SYK	−0.201	0.818	0.626	1.069	0.141
NDRG1_pT346	−0.298	0.742	0.564	0.977	0.033
P90RSK	−0.489	0.613	0.388	0.970c	0.037
TIGAR	−0.304	0.738	0.604	0.903	0.003
XBP1	0.749	2.116	1.077	4.155	0.030

## Data Availability

The protein expression data of GC patients (level 4 data) were freely downloaded from the TCPA database (https://tcpaportal.org/tcpa/ (accessed on 5 March 2022)). The clinical information of these patients was obtained from the TCGA database (https://gdc.cancer.gov/ (accessed on 8 March 2022)).

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
