# Peer review of "Identification of a Novel Protein-Based Prognostic Model in Gastric Cancers"

_biomedicines, 2023, doi:10.3390/biomedicines11030983_

Round 1
Reviewer 1 Report
(1) In Table 1, on first row, "Kaplan-Meier" could be converted to "Kaplan-Meier (P value)" for better understanding.
(2) In line 139, "Establishment" could be converted to "Evaluation"
(3) Figure 4B (survival ROC curve) might not be necessary, because Figure 4C and 4D are enough for analysis with other clinical features.
(4) For Figure 4C and 4D, more detailed descriptions for clinical features (grade, stage, T, N, M) regards to Hazard ratio should be needed. (For example, Hazard ratio of Stage III or II compared to Stage I)
Author Response
First of all, I would like to thank you for taking the time to review our manuscript and give valuable comments to improve our manuscript quality. In addition, we have our manuscript polished by the MDPI Editing Service. We have taken full consideration of your critiques seriously and revised the manuscript accordingly, with the changes highlighted or marked in red in the text. Following are the point-by-point responses to the comments.
Point 1: In Table 1, on first row, "Kaplan-Meier" could be converted to "Kaplan-Meier (P value)" for better understanding.
Response 1: Thank you for your suggestion. We have converted "Kaplan-Meier" to "Kaplan-Meier (P value) " on the first row in Table 1 at your suggestion.
Point 2: In line 139, "Establishment" could be converted to "Evaluation"
Response 2: Thank you for your suggestion. We have converted " Establishment " to "Evaluation" in line 153 at your direction.
Point 3: Figure 4B (survival ROC curve) might not be necessary, because Figure 4C and 4D are enough for analysis with other clinical features.
Response 3: Thank you for your suggestion. We agree with you and have deleted Figure 4B. In addition, we have conducted a time-dependent ROC curve analysis to evaluate the predictive accuracy of the mode according to another reviewer's suggestion.
Point 4: For Figure 4C and 4D, more detailed descriptions for clinical features (grade, stage, T, N, M) regards to Hazard ratio should be needed. (For example, Hazard ratio of Stage III or II compared to Stage I)
Response 4: Thank you for your suggestion. We recognize that it is a crucial issue needed to be considered. According to the reviewer's suggestion, we have made corrections to the revised figures.
Age <60 or ≥60
Gender Male or Female
Grade G1 or G2-3
Stage Stage I-II or Stage III-IV
T Stage T1-2 or T3-4
N Stage N0 or N1-3
M Stage M0 or M1
Reviewer 2 Report
The authors proposed a protein-based model to predict the overall survival of gastric cancer patients. The idea is interesting, and the methods sound appropriate. Nonetheless, a very similar article is already published by Zheng GL, et al. (2022. Front. Oncol. 12:901182. doi: 10.3389/fonc.2022.901182). Below, please find more comments.
Comments:
1. Lines 37-39: the authors state that “… it is reported that the 5-year survival rate of GC patients dropped from 67% to 31% in the United States [2,3]”. I argue against this statement as Rawla, P. et al. (Reference No. 2 of the present article) have written that “the 5-year survival rate for gastric cancer is 31% in the United States. … The 5-year survival rate for pre-metastatic diagnosis is 67%”. I think a misunderstanding of the percentages 31% and 67% has happened.
2. The present study is very similar to “Zheng GL, et al. 2022. Screening Protein Prognostic Biomarkers for Stomach Adenocarcinoma Based on The Cancer Proteome Atlas. Front. Oncol. 12:901182. doi: 10.3389/fonc.2022.901182” in terms of the data and the overall bioinformatics methods used, to such a degree that the figure 1 in the present article is identical to figure 2 in the Zheng GL, et al’s paper. It is recommended to mention the differences between these two analyses and discuss the strengths and shortcomings of the present study in comparison to the Zheng GL, et al's investigation.
3. The manuscript lacks information on the overall characteristics of the subjects (age, gender, tumor type, T, N, M, etc.), the source of data (in materials and methods), and ethical issues declaration.
4. The authors state that their model is validated (e. g. lines 176-177, and 179). However, in my opinion, the proposed model is not still validated because its efficiency was not verified using an independent validation group or with any experimental studies.
5. Table 2: the p-value for SYK protein is not significant. I wonder how a nonsignificant variable can still stay in the model.
6. Line 181: I think it is better to change the “risk factor” to “prognostic factor”.
7. According to Figure 3, lower TIGAR protein expression is associated with the high-risk group. However, according to the discussion, the available literature is in favor of better survival with lower TIGAR levels. This contrast between the available evidence and the present finding is not discussed.
8. Figure 5 remained undiscussed.
Author Response
First of all, I would like to thank you for taking the time to review our manuscript and give valuable comments to improve our manuscript quality. In addition, we have our manuscript polished by the MDPI Editing Service. We have taken full consideration of your critiques seriously and revised the manuscript accordingly, with the changes highlighted or marked in red in the text. Following are the point-by-point responses to the comments.
Point 1: Lines 37-39: the authors state that “… it is reported that the 5-year survival rate of GC patients dropped from 67% to 31% in the United States [2,3]”. I argue against this statement as Rawla, P. et al. (Reference No. 2 of the present article) have written that “the 5-year survival rate for gastric cancer is 31% in the United States. … The 5-year survival rate for pre-metastatic diagnosis is 67%”. I think a misunderstanding of the percentages 31% and 67% has happened
Response 1: Thank you for your kind suggestion. We are very sorry for our misrepresentation. It should be “it is reported that the 5-year survival rate of GC patients was as low as 31% in the United States [2,3]”. We have revised the writing in the Introduction section.
Point 2: The present study is very similar to “Zheng GL, et al. 2022. Screening Protein Prognostic Biomarkers for Stomach Adenocarcinoma Based on The Cancer Proteome Atlas. Front. Oncol. 12:901182. doi: 10.3389/fonc.2022.901182” in terms of the data and the overall bioinformatics methods used, to such a degree that the figure 1 in the present article is identical to figure 2 in the Zheng GL, et al’s paper. It is recommended to mention the differences between these two analyses and discuss the strengths and shortcomings of the present study in comparison to the Zheng GL, et al's investigation.
Response 2: Thank you for your suggestion. The data in both our study and Zheng GL, et al were from public free databases TCGA and TCPA, so the data were consistent and the bioinformatics analysis adopted was identical to some extent. We have deleted Figure 1. Compared with Zheng GL, et al, the advantages of our study were the sample size used for model construction was twice that of Zheng GL, et al, which would be conducive to the credibility of the model. In addition, our evaluation of the model was more reliable as we used the time-dependent ROC to assess the prognostic predictive accuracy of the risk model. Meanwhile, the disadvantage of our study was that model verification was not conducted. In the future, more and more independent cohorts were needed to confirm the feasibility of this model. We have made corrections according to the reviewer's suggestion. The new information was added in the discussion part.
Point 3: The manuscript lacks information on the overall characteristics of the subjects (age, gender, tumor type, T, N, M, etc.), the source of data (in materials and methods), and ethical issues declaration.
Response 3: We thank the reviewer to raise this important issue. According to the reviewer's suggestion, we have made corrections. The new information was added to the revised Materials and Methods and the Results part (Supplemental Table 1). The ethics approval was not required, because all the protein expression data and clinical information of patients in the study were downloaded from public databases, including TCGA (https://tcpaportal.org/tcpa/) and TCPA database (https://gdc.cancer.gov/) and these public databases allowed researchers to download and analyze public datasets for scientific purposes.
Point 4: The authors state that their model is validated (e. g. lines 176-177, and 179). However, in my opinion, the proposed model is not still validated because its efficiency was not verified using an independent validation group or with any experimental studies
Response 4: We are deeply impressed by the reviewer's observation and good suggestions. We were apologized for our misapprehensive and truly agreed with you, this model was not validated. The limitation of our study was the lack of an independent validation group and clinical samples to verify the expression of the five proteins and relevant experimental studies. Further investigations were needed to confirm the feasibility of this model.
Point 5: Table 2: the p-value for SYK protein is not significant. I wonder how a nonsignificant variable can still stay in the model.
Response 5: The SYK protein reserved in the model had been performed with both KM and univariate Cox analyses and it was significant for both analyses. We tried to search the literature to find the reason. Interestingly, we found that the p-value of proteins included in risk model in many studies was not significant. For example, the p-value of COLLAGEN VI and TIGAR in the tri-proteins risk model for stomach adenocarcinoma was not significant [1], and the p-value of EPPK1 in the protein risk model for endometrial carcinoma was also not significant [2]. This issue needs more investigation and discussion in the future.
[1]Zheng GL, Zhang GJ, Zhao Y, Zheng ZC. Screening Protein Prognostic Biomarkers for Stomach Adenocarcinoma Based on The Cancer Proteome Atlas. Front Oncol. 2022 Apr 28;12:901182.
[2]Lai J, Xu T, Yang H. Protein-based prognostic signature for predicting the survival and immunotherapeutic efficiency of endometrial carcinoma. BMC Cancer. 2022 Mar 25;22(1):325.
Point 6: Line 181: I think it is better to change the “risk factor” to “prognostic factor”.
Response 6: Thank you for your suggestion, we have made corrections according to your suggestion.
Point 7: According to Figure 3, lower TIGAR protein expression is associated with the high-risk group. However, according to the discussion, the available literature is in favor of better survival with lower TIGAR levels. This contrast between the available evidence and the present finding is not discussed.
Response 7: Thank you for your suggestion, we have discussed it and the new information was added in the discussion part.
Point 8: Figure 5 remained undiscussed.
Response 8: Thank you for your suggestion. We have added the discussion content at your suggestion.
Reviewer 3 Report
The authors reported “Identification of a Novel Protein-Based Prognostic Model in
Gastric Cancers”. The authors concluded that the robust protein-based model based on 5 proteins was established and can bring potential benefits in the prognostic prediction of gastric patients.
However, there are many problems in this paper, I think.
1. There are many grammatical mistakes in this manuscripts
Ex 1) multivariate cox analysis → Cox
Ex 2) prognosis factors → prognostic factors
Ex 3) Reverse-phase protein arrays (RPPAs) are a powerful proteomic tool → is
2. This paper is not reader-friendly, I think. The authors should the descriptive statistics of patients, tumor characteristics or therapy in Results Part.
3. I think the sample size of this cohort is small to evaluate results.
4. What is the endpoint of this risk model, overall survival (OS) or disease-specific survival (DSS)?
5. If the endpoint is OS, I cannot understand age has a little prognostic significance in univariate, multivariate, and ROC analysis. If the endpoint is DSS, I cannot understand T/N lost prognostic significance in multivariate analysis. If above analysis was correctly performed, this study cohort may be highly biased.
6. What is the clinical value of results of this study in the first place? I do not think this risk model is not for routine clinical practice. Can we change the regimen of chemotherapy obtaining the prediction based on this risk model? The authors should discuss this point.
7. The authors select AUC of ROC when evaluating prognostic predictive accuracy of risk model. Harrell’s concordance index (time-dependent AUC) is better because the authors use Cox model when constructing risk model.
Author Response
First of all, I would like to thank you for taking the time to review our manuscript and give valuable comments to improve our manuscript quality. We have taken full consideration of their critiques seriously and revised the manuscript accordingly, with the changes highlighted or marked in red in the text. Following are the point-by-point responses to the comments.
Point 1: There are many grammatical mistakes in this manuscripts
Ex 1) multivariate cox analysis → Cox
Ex 2) prognosis factors → prognostic factors
Ex 3) Reverse-phase protein arrays (RPPAs) are a powerful proteomic tool → is
Response 1: We apologize for giving you trouble because of those mistakes. At your suggestion, we have made corresponding corrections. In addition, we have our manuscript polished by the MDPI Editing Service.
Point 2: This paper is not reader-friendly, I think. The authors should the descriptive statistics of patients, tumor characteristics or therapy in Results Part.
Response 2: We thank you to figure out this important issue. At your suggestion, we have made corrections according. The new information was added to the revised Results Part (Supplemental Table 1).
Point 3: I think the sample size of this cohort is small to evaluate results.
Response 3: Thanks for your suggestion. I am so sorry that the sample size of this cohort is relatively small. In our study, we have included all the patients with gastric adenocarcinoma in the TCGA database. After removing the missing and unknown data on survival time and status, finally, 352 patients were included in this study to construct the model. In the future, more and more cohorts with large samples would be needed to better construct and evaluate the prognostic model for GC patients.
Point 4: What is the endpoint of this risk model, overall survival (OS) or disease-specific survival (DSS)?
Response 4: The endpoint of this risk model is overall survival (OS).
Point 5: If the endpoint is OS, I cannot understand age has a little prognostic significance in univariate, multivariate, and ROC analysis. If the endpoint is DSS, I cannot understand T/N lost prognostic significance in multivariate analysis. If above analysis was correctly performed, this study cohort may be highly biased.
Response 5: Thanks. To figure out this issue, we have reviewed the literature and found that more and more recent clinical studies have revealed that age was a prognostic factor in patients with GC. For example, Abdulaziz Alshehri et al have reported that older age could independently predict poor overall survival among patients with advanced gastric cancer [1]. Do Dam Suh et al have reported that age was an independent prognostic factor for overall survival among patients with early gastric cancer [2].
[1]Alshehri A, Alanezi H, Kim BS. Prognosis factors of advanced gastric cancer according to sex and age. World J Clin Cases. 2020 May 6;8(9):1608-1619.
[2]Suh DD, Oh ST, Yook JH, Kim BS, Kim BS. Differences in the prognosis of early gastric cancer according to sex and age. Therap Adv Gastroenterol. 2017 Feb;10(2):219-229.
Point 6: What is the clinical value of results of this study in the first place? I do not think this risk model is not for routine clinical practice. Can we change the regimen of chemotherapy obtaining the prediction based on this risk model? The authors should discuss this point.
Response 6: Thanks for your good suggestion. According to your suggestion, we have tried to construct the risk model based on the regimen of chemotherapy. Unfortunately, we found that most of the chemotherapy data were missing from the cohort, and we can not make any conclusion. But, the construction of a risk model based on chemotherapy is a novel and meaningful direction for clinicians. In the future, we will try to construct this model.
Point 7: The authors select AUC of ROC when evaluating prognostic predictive accuracy of risk model. Harrell’s concordance index (time-dependent AUC) is better because the authors use Cox model when constructing risk model.
Response 7: We really appreciate your good suggestion, which is helpful to improve the quality of our manuscript. At your suggestion, we have made corrections according. The time-dependent AUC was added to the revised figure 3 (figure 3B, 3C, 3D, 3E).
Round 2
Reviewer 2 Report
I would like to thank the authors for revising the manuscript. In my opinion, it is improved.
Regarding the authors' response to my previous comment 2: Zheng GL, et al state that “The predictive power of the prognostic risk model was assessed using a Time-dependent ROC curve”. Therefore, I am afraid that the use of this statistical method is not an advantage over Zheng GL et al. paper.
Regarding the authors’ response to my previous comment 4: Lack of a validation set is a major shortcoming in computational research. It is recommended to validate the model on an independent dataset.